# Cell–Fibronectin Interactions and Actomyosin Contractility Regulate the Segmentation Clock and Spatio-Temporal Somite Cleft Formation during Chick Embryo Somitogenesis

**DOI:** 10.3390/cells11132003

**Published:** 2022-06-22

**Authors:** Patrícia Gomes de Almeida, Pedro Rifes, Ana P. Martins-Jesus, Gonçalo G. Pinheiro, Raquel P. Andrade, Sólveig Thorsteinsdóttir

**Affiliations:** 1cE3c—CHANGE, Departmento de Biologia Animal, Faculdade de Ciências, Universidade de Lisboa, 1740-016 Lisboa, Portugal; pgalmeida@fc.ul.pt (P.G.d.A.); pedro.rifes@sund.ku.dk (P.R.); ggdspinheiro@gmail.com (G.G.P.); 2ABC-RI, Algarve Biomedical Center Research Institute, 8005-139 Faro, Portugal; a.patjesus@gmail.com (A.P.M.-J.); rgandrade@ualg.pt (R.P.A.); 3Faculdade de Medicina e Ciências Biomédicas (FMCB), Universidade do Algarve, Campus de Gambelas, 8005-139 Faro, Portugal; 4Champalimaud Research Program, Champalimaud Center for the Unknown, 1400-038 Lisboa, Portugal

**Keywords:** fibronectin, fibronectin matrix assembly, actomyosin contractility, somitogenesis, segmentation clock, *hairy1*, cleft formation

## Abstract

Fibronectin is essential for somite formation in the vertebrate embryo. Fibronectin matrix assembly starts as cells emerge from the primitive streak and ingress in the unsegmented presomitic mesoderm (PSM). PSM cells undergo cyclic waves of segmentation clock gene expression, followed by Notch-dependent upregulation of *meso1* in the rostral PSM which induces somite cleft formation. However, the relevance of the fibronectin matrix for these molecular processes remains unknown. Here, we assessed the role of the PSM fibronectin matrix in the spatio-temporal regulation of chick embryo somitogenesis by perturbing (1) extracellular fibronectin matrix assembly, (2) integrin–fibronectin binding, (3) Rho-associated protein kinase (ROCK) activity and (4) non-muscle myosin II (NM II) function. We found that integrin–fibronectin engagement and NM II activity are required for cell polarization in the nascent somite. All treatments resulted in defective somitic clefts and significantly perturbed *meso1* and segmentation clock gene expression in the PSM. Importantly, inhibition of actomyosin-mediated contractility increased the period of *hairy1/hes4* oscillations from 90 to 120 min. Together, our work strongly suggests that the fibronectin–integrin–ROCK–NM II axis regulates segmentation clock dynamics and dictates the spatio-temporal localization of somitic clefts.

## 1. Introduction

Fibronectin extracellular matrices (ECMs) play key roles in a variety of tissues during embryonic development [1,2]. Fibronectin matrix assembly must be balanced for tissue homeostasis as the absence of fibronectin is embryonic lethal [3], perturbation of fibronectin matrix assembly or cell–fibronectin binding hampers development [4,5,6,7] and excessive fibronectin assembly can lead to pathological conditions such as fibrosis [8].

Fibronectin matrix assembly is a complex cell-dependent process that requires the engagement and unfolding of globular fibronectin by its major assembly receptor, the α5β1 integrin [9,10,11]. As fibronectin dimers are unfolded, their N-terminal matrix assembly domains are exposed and fibronectin–fibronectin binding occurs, giving rise to viscoelastic fibronectin fibrils [9,10,11]. Integrins are linked to the intracellular actomyosin cytoskeleton via intermediate proteins [12,13]. These adhesion complexes, called integrin adhesomes, allow cells to perceive and respond to changes in their physical surroundings [14]. Signalling events in adhesomes impact the actomyosin cytoskeleton through the phosphorylation of non-muscle myosin II (NM II), which binds to actin and converts ATP into mechanical energy [15]. The resulting actomyosin contractility can lead to changes in cell shape, transmit signals from integrin adhesomes to cadherin adhesomes [16] and vice versa, as well as from the cell surface to the nucleus [17,18,19].

One of the most conspicuous morphogenetic events during early vertebrate embryogenesis is the formation of somites. Somites are spheres of epithelioid cells that are formed periodically from the anterior portion of the mesenchymal presomitic mesoderm (PSM), bilateral to the axial structures [20]**,** and are the source of axial skeleton, tendon, dermis and skeletal muscle precursor cells [21]. Periodic somite formation is believed to be regulated by the so-called segmentation clock genes [22,23,24]. These genes are cyclically expressed in the PSM with a period that correlates with the rate of somite formation [24]. As cells reach the anterior third of the PSM, segmentation clock-dependent Notch signalling induces the expression of the transcription factor Mesp2/Meso1 in a band of cells which marks the region where the next somitic cleft will form [25,26,27]. Mesp2/Meso1 then upregulates EphA4, leading to Eph/Ephrin signalling and somitic cleft formation immediately anterior to the Mesp2/Meso1-positive domain [27,28,29,30]. As cells rostral to the cleft epithelialize, a fibronectin matrix is assembled in the forming cleft, and the forming somite undergoes a progressive mesenchymal-to-epithelial transition leading to a centripetal arrangement of polarized epithelioid cells surrounding a mesenchymal core [25,31,32].

Fibronectin is essential for somite formation in all vertebrate models studied to date [3,6,33,34,35,36]. In the chick embryo, primarily ectoderm-derived fibronectin is assembled into a matrix surrounding the α5β1 integrin-expressing PSM tissue [6]. This process starts in the caudal PSM, with the matrix progressively getting denser as the tissue matures anteriorly [37]. This increase in matrix complexity over time correlates with a posterior-to-anterior gradient in cell density [38,39,40] and measurements of the storage modulus of the chick paraxial mesoderm from the tail region to epithelial somites suggest that the tissue progressively gets stiffer as it matures [41,42]. Previous studies have focused on the role of fibronectin in the acquisition of an epithelioid morphology in the rostral PSM and stabilization of the forming clefts [6,31,32,43] and, more recently, on anteroposterior length adjustments of recently formed somites, ensuring left–right symmetry in zebrafish embryos [44]. However, little is known about how the fibronectin matrix regulates PSM maturation over time and the molecular events preceding cleft formation.

In this study, we addressed the involvement of cell–fibronectin interactions and actomyosin contractility in the events leading up to the formation of somites in the chick embryo model. First, we experimentally perturbed extracellular fibronectin fibrillogenesis. Then, we inhibited integrin–fibronectin engagement using the RGD peptide which competes with fibronectin for integrin binding, and also perturbs fibronectin matrix assembly. We then blocked actomyosin contractility indirectly, by inhibiting Rho-associated protein kinase (ROCK), or directly, by blocking NM II ATPase activity. We found that these treatments affected cell polarization in the nascent somites in different ways, suggesting that fibronectin engagement via RGD and ROCK-independent NM II activity are key processes for the acquisition of the epithelioid morphology. All treatments lead to defects in somitic cleft formation. Importantly, they also led to alterations in the expression of segmentation clock genes and a mispositioning of the *meso1* expression domain in the rostral PSM, suggesting that the molecular machinery specifying the spatio-temporal location of somite clefts was perturbed. Finally, blocking NM II ATPase activity significantly delayed *hairy1* mRNA oscillations in the PSM, evidencing, for the first time, that actomyosin contractility regulates the pace of the segmentation clock.

## 2. Materials and Methods

### 2.1. Embryos and Experimental Design

Fertilized chicken (*Gallus gallus*) eggs were obtained from commercial sources (Sociedade Agrícola Quinta da Freiria or Pintobar Exploração Avícola, Lda, Portugal) and incubated at 37.5 °C in a humidified chamber until the desired HH stage (HH4 or HH11-14; [45]). Somite nomenclature is according to [46].

To evaluate the requirement of cell–fibronectin interactions and actomyosin contractility for somite formation, we interfered with the fibronectin–integrin–actomyosin axis at different levels using four experimental treatments (Figure 1A): (1) over-expression of the 70 kDa fibronectin fragment, a dominant-negative inhibitor of fibronectin matrix assembly [47,48] (Figure 1B); (2) incubation with a linear RGD peptide which competes with fibronectin for integrin binding [49,50] (Figure 1C); (3) incubation with RockOut, a chemical inhibitor of ROCK I and ROCK II enzymes involved in activating NM II [51] (Figure 1C) and (4) incubation with Blebbistatin, which directly inhibits the ATPase activity of NM II and consequently, actomyosin contractility [52] (Figure 1C).

### 2.2. Embryo Electroporation and ex ovo Culture

HH4-5 embryos were electroporated on one (randomly selected) side of the primitive streak in the presumptive PSM and/or ectoderm and cultured ex ovo using the Early Chick culture method [53] (Figure 1B). The electroporation mixture contained plasmid DNA at 0.5–1 µg/µL mixed with 0.4% Fast Green for visualization. Embryos were submerged in an electroporation chamber filled with Tyrode’s saline and three pulses of 6–9 V, 50 ms each, at 350 ms intervals were applied. Control embryos were electroporated with pCAGGs containing a GFP reporter (pCAGGs-GFP; abbreviated pCAGGs) [54]. pCAGGs-70 kDa qFN1 was kindly provided by Yuki Sato [48] and was co-electroporated with the pCAGGs-GFP plasmid in experimental embryos (treatment abbreviated 70 kDa). Electroporated embryos were screened for GFP after fixation to select embryos with an intense signal on only one side for further analyses.

### 2.3. Embryo Explant Culture and Treatments

A tissue explant culture system [24] was used to assess the effect of the RGD peptide, RockOut and Blebbistatin on somitogenesis (Figure 1C). HH11-14 embryos were collected, bisected along the midline and then cut transversally rostral to somites IV and Hensen’s node. The two contralateral halves thus retained half of the neural tube and notochord, as well as the first four somites and the PSM, with all remaining neighbouring tissues intact. The contralateral explant halves were placed in two different wells, on a polycarbonate filter floating on M199 medium supplemented with 10% chick serum, 5% foetal calf serum and 100 U/mL of penicillin and streptomycin and cultured at 37 °C with 5% CO_2_ [24]. The linear RGD peptide (GRGDS–G4391, Sigma, Lisbon, Portugal) was diluted in culture medium and used at 0.9 mM, while control explants were cultured in medium only. RGD peptide efficiency was confirmed in a cell adhesion assay [50,55] before using it on explants. RockOut (Calbiochem, Gibbstown, NJ, USA) and InSolution™Blebbistatin (Calbiochem) in DMSO were used at a final concentration of 50 µM in culture medium. Equal volumes of DMSO (Sigma) were used as control for both drugs.

### 2.4. Immunohistochemistry

Cryosectioning was performed on embryo explants and whole embryos fixed in 4% paraformaldehyde in 0.12 M phosphate buffer containing 4% sucrose. Fixed samples were washed in 0.12 M phosphate buffer with 4% and 15% sucrose and then embedded in 7.5% gelatine in 0.12 M phosphate buffer containing 15% sucrose, frozen on dry-ice-chilled isopentane and stored at −80 °C until sectioning. Cryostat sections (10–30 µm) were processed for immunofluorescence as previously described [56]. Permeabilization of sections was performed with 0.2% Triton-X100 in phosphate buffered saline (PBS). For blocking, 5% bovine serum albumen (BSA) or a combination of 1% BSA and 10% normal goat serum in PBS were used depending on the presence or absence, respectively, of anti-fibronectin antibodies. Primary and secondary antibodies were diluted in 1% BSA in PBS. Sections were incubated with primary antibodies overnight at 4 °C and with secondary antibodies for 1 h at room temperature.

For whole-mount immunodetection, explants were fixed in 4% paraformaldehyde in PBS and processed as previously described [31,37]. An amount of 1% Triton-X100 in PBS was used for permeabilization and 1% BSA in PBS was used for blocking and antibody dilution. Both primary and secondary antibody incubations were performed overnight at 4 °C.

The following primary antibodies were used: anti-ZO-1 (Zymed, Waltham, MA, USA, #40-2200, 1:100 or Invitrogen, Waltham, MA, USA, #33-9100, 1:100); anti-*N*-cadherin (BD Biosciences, Franklin Lakes, NJ, USA, #610920, 1:100); anti-fibronectin (Sigma, #F-3648, 1:400), anti-activated caspase3 (Cell Signaling, Danvers, MA, USA, #9661, 1:1000) and anti-GFP (Invitrogen, #A11122, 1:100). For staining DNA we used ToPro3 (Invitrogen, 1:500) in conjunction with ribonuclease A (Sigma, 10 µg/mL), 4% Methyl Green (Sigma, diluted 1:250; [57]) or 4′,6-diamidino-2-phenylindole (DAPI, 5 µg/mL in PBS with 0.1% Triton-X100). For detection of the primary antibodies the adequate secondary goat anti-mouse and anti-rabbit Alexa 488-, Alexa 568- or Alexa 546-conjugated F’ab fragments from Invitrogen were used (#A-11017, #A-21069, #A-11071, #A-11019, #A-11070, 1:1000). Immunohistochemistry was performed on at least 6 different embryos/explants and the respective controls for each treatment (70 kDa *n* = 7; pCAGGs *n* = 6; RGD *n* = 13; RockOut *n* = 15; Blebbistatin *n* = 13).

### 2.5. In Situ Hybridization

In situ hybridization using DIG-labelled RNA probes was performed as described previously [58] with minor alterations [56]. RNA probes were synthetized from linearized plasmids: *meso1* [59], *hairy1* [24] and *hairy2* [60]. Only intact 70 kDa-electroporated embryos (without detached tissues) were processed for in situ hybridization.

### 2.6. Sample Preparation and Imaging

Cryostat sections were mounted in Vectashield (Vector Laboratories) or in 5 mg/mL propyl gallate in glycerol/PBS (9:1) with 0.01% azide. Whole mount explants were gradually dehydrated in methanol and cleared in methyl salicylate (Sigma) as described previously [31,37]. Immunofluorescence images were taken on a confocal Leica SPE microscope, following imaging acquisition steps described previously [37]. Image analysis was performed using Fiji v. 1.49 (https://imagej.net/Fiji, accessed on 15 June 2022) software. Image histogram corrections and, when appropriate, maximum intensity projections of immunofluorescence confocal stacks were produced in Fiji. When applicable, contiguous images were stitched together into a single image using the pairwise stitching Fiji plugin [61]. Image acquisition of embryos and explants processed for in situ hybridization was performed using a Zeiss LUMAR V12 Stereoscope coupled to a Zeiss Axiocam 503 colour 3MP camera. Imaging of explants prior to in situ hybridization was performed in 50% formamide solution. For comparing the in situ hybridization patterns along the paraxial mesoderm, the Fiji plugin Straighten [62] was used and contralateral explant pairs were aligned by SIV.

### 2.7. Statistical Analysis

Paired Student’s *t*-tests were performed to assess for differences in the number of somites formed in embryos electroporated with pCAGGs only and pCAGGs + 70 kDa and in RGD-, RockOut- and Blebbistatin-treated explants relative to the respective controls. Differences in the frequency of morphological and gene expression phenotypes found in 70 kDa-electroporated embryos compared to pCAGGs-electroporated control embryos was tested through a chi-square test. Differences between the number of *hairy1* expression stripes formed in explants cultured with DMSO or Blebbistatin for the same amount of time was performed using a Kruskal–Wallis test. Statistical significance was set at *p* < 0.05. Statistical analyses were performed in Statistica 10 (https://statistica.software.informer.com/10.0, accessed on 15 June 2022) and Graphpad Prism 5 (https://graphpad-prism.software.informer.com/5.0/, accessed on 15 June 2022).

## 3. Results

### 3.1. Interfering with Fibronectin Matrix Assembly, Cell–Fibronectin Binding or Actomyosin Contractility Leads to Defective Somite Cleft Formation

Four experimental treatments were used to interfere with the fibronectin–integrin–actomyosin axis (Figure 1A), none of which led to an increase in apoptosis (Appendix A). To inhibit fibronectin matrix assembly, a plasmid expressing the 70 kDa fragment was delivered by electroporation to one side of the primitive streak of HH4-5 embryos, which were then re-incubated for 26 h (Figure 1B). The contralateral non-electroporated tissue, as well as embryos electroporated only with the pCAGGs plasmid, were used as controls. Embryos electroporated with the pCAGGs vector alone formed, on average, 14.5 pairs of somites (Figure 2A), which was as expected for these stages [45]. The 70 kDa-electroporated embryos displayed an array of somitic phenotypes with variable severity (Appendix A). Although 70 kDa-electroporated embryos formed, on average, the same number of somitic structures (Figure 2A), these were very ill-defined when compared to the somites of control embryos (arrowheads in Figure 2B), lacking a clear delimitation of their boundaries. In some cases (15/144) somite-like clefts were no longer discernible (posterior tissue in Figure 2B) and fused somitic structures could occasionally be observed (Figure 2C). The 70 kDa-electroporated embryos could also display a kinked neural tube (Appendix A) and detached tissues (Appendix A), which are reminiscent of the phenotypes obtained after interfering with fibronectin matrix deposition and/or with fibronectin–integrin binding [3,63,64,65,66].

The fibronectin matrix of pCAGGs-electroporated embryos was thick and located near the surface of the paraxial mesoderm (white arrowheads in Figure 2D). In contrast, the fibronectin matrix of 70 kDa-electroporated embryos was thinner and disrupted (open arrowheads in Figure 2E,F), confirming that the 70 kDa fragment impaired fibronectin matrix assembly in our system. The ectoderm and endoderm were often separated from the paraxial mesoderm in 70 kDa-electroporated embryos (brackets in Figure 2E’,F’ and Appendix A), suggesting that the fibronectin matrix was weakened to the point of being insufficient to hold these tissues together.

We then focused on the morphology of the most recently formed somites. Figure 2D,D’ shows a young SI somite (control), that has formed its anterior and posterior clefts (asterisks in Figure 2D’) but has not yet completed the epithelialization of its anterior side [31], as seen by the absence of ZO-1 staining in this region (Figure 2D). The somitic cells of both mildly (Figure 2E) and severely affected (Figure 2F) 70 kDa-electroporated embryos accumulated ZO-1 apically in a similar pattern to that of controls (Figure 2D). However, they displayed incomplete clefts (arrows in Figure 2E’,F’,H). This phenotype ranged from giving rise to fused somitic structures (Figure 2C) to embryos where no complete cleft was formed, originating a string of connected somitic structures (Figure 2H). This was never observed in control embryos (Figure 2G). Altogether, these results indicate that an intact fibronectin matrix is required for proper somite cleft formation.

Posterior embryo half explants cultured with the RGD peptide for 6 h (Figure 1C) formed slightly fewer somite-like structures than the contralateral controls, a difference that was statistically significant (Figure 3A). This was also true in explants cultured with RockOut for 6 h, which only formed an average of 3.1 somitic structures, compared to 3.6 in the DMSO control (Figure 3B). Importantly, the somite-like structures formed in the presence of RGD or RockOut were often delimited by poorly defined clefts (Appendix A), when compared to control somites, similarly to what was observed in 70 kDa-electroporated embryos (Figure 2B). Finally, explants cultured with Blebbistatin formed an average of 1.5 somitic structures during the 6 h incubation period (Figure 3B), and these were always ill-defined (Appendix A).

We next analysed somite morphology after 6 h of culture (Figure 3D–I, Figure 4B–G and Appendix A), focusing on two different axial levels: S0 level at the end of culture (an area that was at stage S-IV at the beginning of the culture; red box in Figure 3C) and at SII/SIII level at the end of culture (an area that was at stage S-II/S-I at the beginning of the culture, i.e., in the determined region of the PSM, [67]; red box in Figure 4A). Control S0 somites presented centripetally organized nuclei, were clearly separated from SI (asterisks in Figure 3D’,F’,H’) and had assembled fibronectin within the cleft separating these two somites (yellow arrowheads in Figure 3D,F,H). The S0 somites of the contralateral explants cultured in the presence of RGD or RockOut were not fully separated from SI (arrows in Figure 3E’,G’) and lacked assembled fibronectin where the cleft should be (yellow open arrows in Figure 3E,G). Additionally, in RGD-treated explants the S0 nuclei were not centripetally organized (Figure 3E’), nor was *N*-cadherin polarization complete (Appendix A) compared to the control side (Appendix A). The polarization of somitic cells in RockOut-treated explants occurred normally (Figure 3F’,G’ and Appendix A). The most severe defects were observed in explants cultured with Blebbistatin, which inhibits NM II directly. Here, only one or two ill-defined somitic structures were formed (Figure 3B,I,I’). A sagittal section of the tissue where somite S0 should have formed did not show any signs of cleft formation, centripetally organized nuclei (Figure 3H’,I’ and Appendix A) nor *N*-cadherin polarization (Appendix A), while epithelial tissues other than somites (e.g., ectoderm and neural tube) retained cellular organization and *N*-cadherin polarization after incubation (Appendix A). Even when the explants were allowed to develop for an additional 4.5 h, no additional somite-like structures were formed in Blebbistatin-treated explants (Figure 3B). This was in contrast to explants cultured in RGD or RockOut, which continued to form somites (Figure 3A,B). We conclude that PSM tissue that was at stage S-IV when explants were exposed to RGD, RockOut or Blebbistatin and then cultured for 6 h (red box in Figure 3C) fails to form a normal cleft. Moreover, explants cultured with RGD or Blebbistatin show an impairment or even a complete block of cell polarization (Appendix A).

Somites SII/SIII in control explants were well separated from each other (asterisks in Figure 4B’,D’,F’) and displayed an assembled fibronectin matrix within the clefts between somites (yellow arrowheads in Figure 4B,D,F). These somites in RGD-treated explants had polarized *N*-cadherin (Appendix A), but cleft formation was incomplete (Figure 4B’,C’; Appendix A) and there was deficient fibronectin matrix accumulation within the cleft (Figure 4B,C). SII/SIII somites in RockOut-treated explants were indistinguishable from the somites of the control side (Figure 4D,D’,E,E’ and Appendix A). In contrast, this tissue in Blebbistatin-treated explants lacked centripetal nuclear alignment (Figure 4F’,G’ and Appendix A), *N*-cadherin polarization (Appendix A) and proper cleft formation (Figure 4F’,G’ and Appendix A). We conclude that PSM tissue that was at stage S-II/S-I (determined PSM; [67]) when explants were exposed to RGD, RockOut or Blebbistatin (red box in Figure 4A) is less perturbed after a 6 h incubation in these experimental conditions than the S-IV PSM area. Nevertheless, cleft formation is impaired in RGD- and Blebbistatin-treated explants and somitic cells in Blebbistatin-treated explants did not polarize (Appendix A).

In summary, electroporation of the 70 kDa-expressing construct and incubation with RGD, RockOut or Blebbistatin affect cell polarization to different extents (Appendix A), but all cause impairment in somitic cleft formation. We conclude that cell–fibronectin interactions and actomyosin contractility are required for somite cleft formation.

### 3.2. Interfering with Fibronectin Matrix Assembly, Cell–Fibronectin Binding or Actomyosin Contractility Alters meso1 and Cyclic hairy1 Expression

We next asked whether the prepatterning mechanisms that determine the correct spatio-temporal appearance of clefts in the rostral PSM was altered in our experimental conditions. Cleft formation is induced by the transcription factor Mesp2/Meso1 which is upregulated by Notch signalling downstream of the segmentation clock [27,68]. In the chick embryo, *meso1* expression is upregulated at the level of S-II, remains expressed until this area becomes S-I and is then downregulated, after which it is upregulated again in S-II and the cycle repeats [59]. Electroporation of the 70 kDa fragment or the pCAGGs control resulted in alterations to the localization or intensity of *meso1* expression (Figure 5A–C). Nevertheless, there was a significant increase in the frequency of *meso1* expression alterations in embryos electroporated with the 70 kDa fragment, when compared with controls (*p* < 0.01; Figure 5A–C). In total, 40% of embryo explants cultured in the presence of the RGD peptide for 6 h displayed alterations in *meso1* expression when compared to the contralateral control (Figure 5D,F). We also cultured explant pairs for 9 h, which gave similar results (Figure 5E,F). *Meso1* was expressed either in a different location (Figure 5D) or with a different intensity (Figure 5E) on the RGD-treated side compared to the contralateral control. *Meso1* expression was altered in all RockOut-treated explants cultured for 6 h (Figure 5I), being either weaker/absent (Figure 5G) or in a different location (Figure 5H), compared to the contralateral control explant. Similar results were obtained in Blebbistatin-treated explants cultured for 6 h (Figure 5I–K). Since these treatments affect the expression of *meso1* in S-II, we conclude that they act on the PSM tissue from stage S-VI to S-II (red box; Figure 5L). Moreover, *meso1* expression was already perturbed after 3 h of culture with RockOut or Blebbistatin (*n* = 5/5 and 8/9, respectively; Figure 5M,N), corresponding to an effect between S-IV and S-II (blue box; Figure 5L). Together, these results indicate that perturbing fibronectin matrix assembly, cell–fibronectin interactions or actomyosin contractility significantly alters the cycle of activation and suppression of *meso1* in the rostral PSM. This is consistent with the perturbations observed at the level of somitic cleft formation (described in the previous section).

*Meso1* expression dynamics are dependent on the operation of the segmentation clock [27,68]. We therefore assessed the expression pattern of the segmentation clock gene *hairy1* (*hes4*) [24] in our experimental conditions. The majority of embryos electroporated with the 70 kDa fragment had different *hairy1* expression patterns on the electroporated versus the control side, which was significantly less frequent (*p* < 0.01) in embryos electroporated with the pCAGGs vector alone (Figure 6A–C). Similar results (Figure 6D–F; *p* < 0.001) were obtained when analysing the expression of *hairy2*, another segmentation clock gene [60]. The *hairy1* expression patterns in explants cultured with RGD (Figure 6G–I), RockOut or Blebbistatin (Figure 6J–L) were different from the contralateral control sides in 60%–64% (RGD) and 78%–80% (RockOut and Blebbistatin) of the cases. This was reminiscent of the alterations in *meso1* expression (Figure 5), suggesting that perturbed segmentation clock dynamics underly the observed alterations in *meso1* expression patterns. Importantly, our results suggest that segmentation clock oscillations in the chick PSM are dependent on fibronectin matrix assembly, cell–fibronectin interactions and actomyosin contractility.

### 3.3. Delayed Periodicity of hairy1 Oscillations in Explants Treated with Blebbistatin

We next questioned the nature of the perturbation to *hairy1* expression caused by our experimental treatments. Different expression patterns in treated explants could result from halted expression dynamics (while oscillatory behaviour proceeds in the control explant) or from oscillations with a different periodicity from controls. We addressed this experimentally using explants incubated in the presence of Blebbistatin. The Blebbistatin treatment offered two important advantages for this purpose: (1) a high percentage of explants showed altered expression of *hairy1* (Figure 6L) and (2) Blebbistatin-treated explants formed only 1–2 ill-defined somitic structures (Figure 3B). Since morphological somite formation is uncoupled from molecular segmentation [24,69] and a *hairy1* expression stripe marks the caudal region of each segment [24], it was possible to quantify how many segmentation clock cycles were completed in culture with Blebbistatin by counting the number of *hairy1* expression stripes, without the confounding effect of morphological somite formation.

We cultured bisected contralateral explant pairs separately in medium supplemented with DMSO or with Blebbistatin (Figure 7A). After 6 h of incubation, one of the explants was fixed and the contralateral one was either also fixed (6 h + 0 min; Figure 7A,B) or allowed to develop further for the desired period of time (6 h + 60 min, 6 h + 90 min, 6 h + 120 min; Figure 7A,B), and then both were processed by in situ hybridization for *hairy1*. Contralateral explant pairs fixed at the same time were always in the same phase of the *hairy1* expression cycle (Figure 7C,D’,F’) and formed the same number of *hairy1* expression stripes in the rostral PSM (Figure 7B,D’,F’; Appendix A). Control explants formed an average of 4.05 *hairy1* expression stripes in 6 h, as expected (Figure 7B), with corresponding somite formation (Appendix A). In contrast, Blebbistatin-treated explants formed an average of 2.96 *hairy1* stripes (Figure 7B). This difference was statistically significant (*p* < 0.001; Figure 7B) and suggests that molecular segmentation is delayed in Blebbistatin-treated explants compared to DMSO-explants.

We then analysed explant pairs where one of the explants was cultured for an additional 60, 90 or 120 min before fixation and in situ hybridization for *hairy1*. As expected, all control explant pairs with a difference of incubation time of 60 min presented different phases of the *hairy1* expression cycle (Figure 7C,E’; Appendix A). This was also true for Blebbistatin-treated explants (Figure 7C,G’; Appendix A), evidencing that inhibition of the actomyosin contractility does not stop segmentation clock oscillations. When control explants were cultured for 6 h + 90 min, they all presented an additional *hairy1* stripe and were all in the same phase of the *hairy1* expression cycle as their contralateral half fixed 90 min earlier (Figure 7C,H’; Appendix A). This result is consistent with the 90 min period of the segmentation clock in the chick embryo [24]. In contrast, only half (*n* = 2/4) of the explants cultured for an additional 90 min in Blebbistatin-containing medium formed another *hairy1* stripe (Figure 7B; Appendix A). Importantly, all explant pairs were in a different phase of the *hairy1* expression cycle (Figure 7C,J’; Appendix A). These results clearly show that although the *hairy1* expression pattern changes within 90 min in the presence of Blebbistatin, it does not complete a full cycle.

The observation that Blebbistatin-treated explants formed three *hairy1* stripes in 6 h (Figure 7B) suggested that each stripe may take 2 h to form. Hence, we analysed explant pairs differing in 120 min of incubation in control and Blebbistatin-containing medium. Control explants incubated for another 120 min had one more *hairy1* stripe and were in a different phase of the *hairy1* expression cycle compared to their contralateral pair, since they completed one *hairy1* cycle (after 90 min) and had started the next one (Figure 7B,C,I’; Appendix A). Remarkably, explants cultured in Blebbistatin for an additional 120 min not only presented a new molecular stripe but were also in the same phase of the *hairy1* expression cycle as their contralateral pairs, fixed 120 min earlier (Figure 7B,C,K’; Appendix A). This indicates that in the presence of Blebbistatin, *hairy1* expression oscillates with a periodicity of 120 min.

In summary, the average number of *hairy1* stripes increased over time in both DMSO- and Blebbistatin-treated explants (grey/purple trendlines, Figure 7B) and this increase was statistically significant for both DMSO (*p* = 0.02) and Blebbistatin (*p* = 0.05). However, the recapitulation of *hairy1* expression with the addition of a new molecular segment occurred after 90 min in control explants, while this took 120 min in the presence of Blebbistatin. We conclude that Blebbistatin does not block cyclic *hairy1* expression nor PSM molecular segmentation, but that it slows down the period of the cycle to 120 min resulting in the formation of one molecular segment every 2 h. Hence, actomyosin contractility regulates the period of the somitogenesis clock in the chick PSM.

## 4. Discussion

### 4.1. Integrin–Fibronectin Engagement and ROCK-Independent Actomyosin Contractility Are Required for Somite Epithelialization

In this work we assessed the role of cell–fibronectin communication and actomyosin contractility during chick somitogenesis using experimental perturbations targeting (1) fibronectin matrix assembly, (2) fibronectin–integrin interaction, (3) ROCK I/II and (4) NM II activities. All treatments lead to defects in somite cleft formation. Our results confirmed the reported role of the fibronectin matrix in cell polarization in the forming somite and in the stabilization of the nascent cleft [6,31,32,43]. However, the treatments affected cell polarization in the nascent somites in different ways, which adds novel insight into how this process is regulated.

The presence of RGD hampered cell polarization in recently formed somites, while perturbing fibronectin matrix assembly with the 70 kDa-construct did not. The reason for this difference is not clear, but while both the 70 kDa construct and addition of RGD perturb fibronectin assembly, as evidenced by these embryos displaying a thinner fibronectin matrix (Figure 2E,F and Figure 3E) compared to controls, only RGD blocks integrin–fibronectin binding. Hence, cell polarization during somite emergence may be more sensitive to integrin engagement with the surrounding fibronectin matrix than to its assembly state. Our findings are supported by the notion that integrin–fibronectin engagement on the basal cell surface leads to signalling via focal adhesion kinase (FAK), which drives cell polarization and the formation of adherens junctions in a variety of cell types [2,17,18]. Consistent with this, studies in mouse, *Xenopus* and zebrafish indicate that α5β1–fibronectin engagement during somite epithelialization activates FAK [32,70,71,72]. A curious observation was that cell polarization was unaltered in more mature SII and SIII somites after RGD incubation (Figure 4B,B’,C,C’). The region of S-II already has polarized *N*-cadherin [31] and this suggests that the determined PSM tissue retains cell polarity even after blocking fibronectin–integrin interactions. An alternative explanation is that once the somite is formed, cell polarity can be restored through an integrin-independent process, as the somite matures. In the future it would be interesting to elucidate which is the case.

Blebbistatin inhibited cell polarization most drastically, with no discernible *N*-cadherin polarization or centripetal alignment of nuclei in S0 and only incipient organization of SII/SIII. In contrast, cell polarization in explants treated with RockOut was mostly unaffected. RockOut targets NM II activity indirectly by inhibiting ROCK I and II, two of the kinases that activate NM II [73]. In contrast, Blebbistatin targets the NM II ATPase directly. Our results thus raise the possibility that the acquisition of cell polarization in nascent somites is independent of ROCK activity, being dependent on another NM II activator. Ca^++^/calmodulin signalling can activate NM II via myosin light chain kinase (MLCK; [74]) and inhibiting calmodulin was shown to block cell polarization during chick somitogenesis [75]. Alternatively, FAK can activate Rac1 downstream of integrin–ECM engagement [18] and Rac1 in turn activates NM II and promotes its localization to cell–ECM contacts [76]. During chick somitogenesis, the levels of Rac1 activation need to be tightly regulated, since both overactivation and inhibition lead to a failure in epithelialization [77]. Further studies are needed to decipher the molecular pathways involved in the polarization of cells in the nascent somite. However, our data are consistent with a model where integrin–fibronectin engagement and ROCK-independent actomyosin contractility are required for the emergence of the epithelial somite.

### 4.2. A Fibronectin–Integrin–ROCK–NM II Axis Contributes to the Prepatterning of the PSM for Cleft Formation by Regulating the Cycles of hairy1 and meso1 Expression

We showed that interfering with the fibronectin–integrin–ROCK–NM II axis at multiple levels results in defective somite clefts. This led us to ask whether the molecular segmentation of the PSM tissue in preparation for cleft formation was also affected. We found that all treatments lead to different patterns of *hairy1* and *meso1* expression on the experimental versus control sides of the same embryo, indicating that fibronectin–cell communication and actomyosin contractility regulate the normal cycles of expression of these two genes in the PSM. In agreement with our results, chicken embryos electroporated with RNAi constructs against integrin β1 showed alterations in *hairy2*, *Lfng* and *meso1* expression in the PSM [78]. Mouse embryos where the fibronectin RGD site was substituted with an RGE sequence (*Fn1^RGE/RGE^*) also showed asymmetric and/or dampened expression of *Lfng* and *Hes7* in the PSM [65], and *EphA4*, a direct target of Mesp2 in the anterior PSM [29], was diffusely expressed or absent [65]. Together with these studies, our results strongly suggest that the fibronectin–integrin–ROCK–NM II axis regulates the dynamics of the segmentation clock with effects on the prepatterning of the PSM tissue for cleft formation.

To better understand the nature of the perturbations in the expression pattern of *hairy1*, we designed experiments to assess how inhibition of actomyosin contractility affected the dynamics of *hairy1* expression in the PSM. Importantly, incubation with Blebbistatin did not stop the oscillatory expression of *hairy1* in the PSM. Rather, *hairy1* dynamics were delayed, oscillating with a period of 120 min instead of 90 min, concomitant with the formation of a new molecular segment (*hairy1* stripe) in the rostral PSM. Hence, we show for the first time that the PSM cells’ actomyosin contractility regulates the period of *hairy1* expression oscillations. Our finding that inhibition of actomyosin contractility at the level of ROCK activity results in the formation of only 3 somitic structures in 6 h is in agreement with this result.

### 4.3. The PSM Cells’ Mechanical Landscape as a New Player in Temporal Control of the Segmentation Clock

It is well established that Notch signalling plays a role in the segmentation clock and is required for timely *meso1* activation [27]. Relevant to our work is that cell mechanics play a crucial role in Notch activation [79,80,81]. Actomyosin contractility influences the capacity of cells to engage with neighbouring cells through close-range lateral contacts as well as through longer-range cell protrusions, both of which are relevant for Notch signalling [82,83]. For example, in the *Drosophila* notum, Notch and Delta are found on basal protrusions where they can signal at a distance [84,85]. Notch signalling was shown to require actomyosin contractility in both signal-sending and -receiving cells, suggesting that basal protrusions of Notch- and Delta-expressing cells both need NM II-dependent tension for robust signalling [85]. In our experiments, the strongest effect on *hairy1* and *meso1* expression was obtained when actomyosin contractility was blocked with either RockOut or Blebbistatin (Figure 5I and Figure 6L). This suggests that the alterations to segmentation clock periodicity may be downstream of perturbed Notch–Delta signalling.

Chick PSM cells are highly motile cells which dynamically extend and retract filopodia [31,38]. In the caudal PSM, these cells undergo random movement downstream of Fgf signalling, giving rise to a caudal to rostral cell motility gradient in the PSM, which drives axis elongation [38]. This motile behaviour requires actomyosin contractility because incubation of chick embryos with a ROCK inhibitor or Blebbistatin reduced cell motility and blocked axis elongation [38]. The caudal PSM tissue is surrounded by a fibronectin matrix [37] and cells express *N*-cadherin on their cell surface since they emerge from the primitive streak [86,87]. Changing this mechanical landscape by interfering with cell–ECM interactions or by blocking actomyosin contractility will, in analogy with the findings in the *Drosophila* notum [85], reduce the frequency and/or stability of cell–cell contacts and therefore affect the efficiency of Notch signalling.

Studies in zebrafish have elucidated the effect of PSM cell–cell communication via Notch signalling on the rate of the somitogenesis clock and somite formation. The transplantation of cells treated with *her1/her7* morpholinos, which causes upregulation and sustained DeltaC expression, into the zebrafish PSM accelerated *her1* oscillations and the rate of segment formation in the host [88]. Liao et al. [89] also observed faster *her1* oscillations and somite segmentation in the zebrafish *Damascus* mutant line, containing a 50-fold excess of DeltaD. In contrast, reduction in Delta–Notch coupling (in Delta–Notch mutants and DAPT-treated embryos) delayed the segmentation clock and the rate of somite formation [90]. Our Blebbistatin-treated explants show similar results. When actomyosin contractility is blocked, the period of *hairy1* oscillations slows down and the PSM gives rise to fewer molecular segments per time unit. Together, these results strongly suggest that softening of the mechanical landscape of PSM cells leads to less efficient or less frequent Notch signalling, as seen by a longer period of *hairy1* oscillations. Importantly, oscillations do not stop, but continue at a different rate. This raises the interesting possibility that the period of the segmentation clock is regulated by the mechanical properties of cells in the PSM tissue, which act to modulate the efficiency of Notch signalling. Future studies should address whether tissue-specific mechanical landscapes underlie the different periods of the segmentation clock observed during the formation of the last somites [91], in the developing forelimb [92] or even in the PSM of different organisms, where it has been shown to depend on the kinetics of biochemical reactions [93,94].

## 5. Conclusions

Our work evidenced that integrin–fibronectin engagement and ROCK-independent actomyosin contractility are required for the emergence of the epithelial somite. Additionally, we showed that interfering with the fibronectin–integrin–ROCK–NM II axis perturbs *hairy1* expression dynamics, spatio-temporal upregulation of *meso1* in the rostral PSM and somite cleft formation. Importantly, our results uncover a previously unappreciated role of the actomyosin contractility of PSM cells in setting the periodicity of segmentation clock gene expression oscillations. The acknowledgment that cell mechanical properties can regulate the pace of gene expression dynamics may significantly impact the study of other cell types and diseases where signalling dynamics and ECM mechanics play critical roles [8,11,95,96].

## Figures and Tables

**Figure 1 cells-11-02003-f001:**
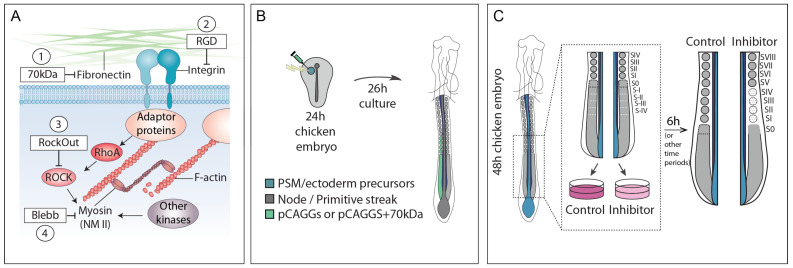
Experimental approaches to challenge the fibronectin–integrin–actomyosin communication axis. (**A**) Schematic representation of the four experimental treatments used: (1) expression of the 70 kDa N-terminal fragment of fibronectin impairs fibronectin matrix assembly; (2) addition of the linear RGD peptide to the culture medium competes with native fibronectin for integrin α5β1 engagement, with effects on fibronectin matrix assembly and cell–fibronectin interactions; (3) addition of RockOut, a chemical inhibitor of ROCK I and II, enzymes involved in activating NM II downstream of integrins; and (4) addition of Blebbistatin, a chemical inhibitor that blocks NM II activation directly. (**B**) Electroporation strategy used for 70 kDa overexpression. PSM/ectoderm progenitors of primitive-streak stage embryos were electroporated on one side with either a pCAGGs GFP-expressing vector alone (pCAGGS), or co-electroporated with pCAGGs-GFP and a 70 kDa-expressing vector (70 kDa). Electroporated embryos were then cultured ex ovo for 26 h using the Early Chick culture method [53], after which they were fixed. (**C**) Incubation of tissue explants with RGD or chemical inhibitors. HH11-14 embryos were collected, and the posterior region was isolated and bisected into two embryo half explants. Control (right or left) explants were cultured in medium only or medium containing DMSO, while experimental (left or right) explants were cultured in the presence of RGD, RockOut or Blebbistatin for the designated time. Both explant halves were fixed at the end of the culture period.

**Figure 2 cells-11-02003-f002:**
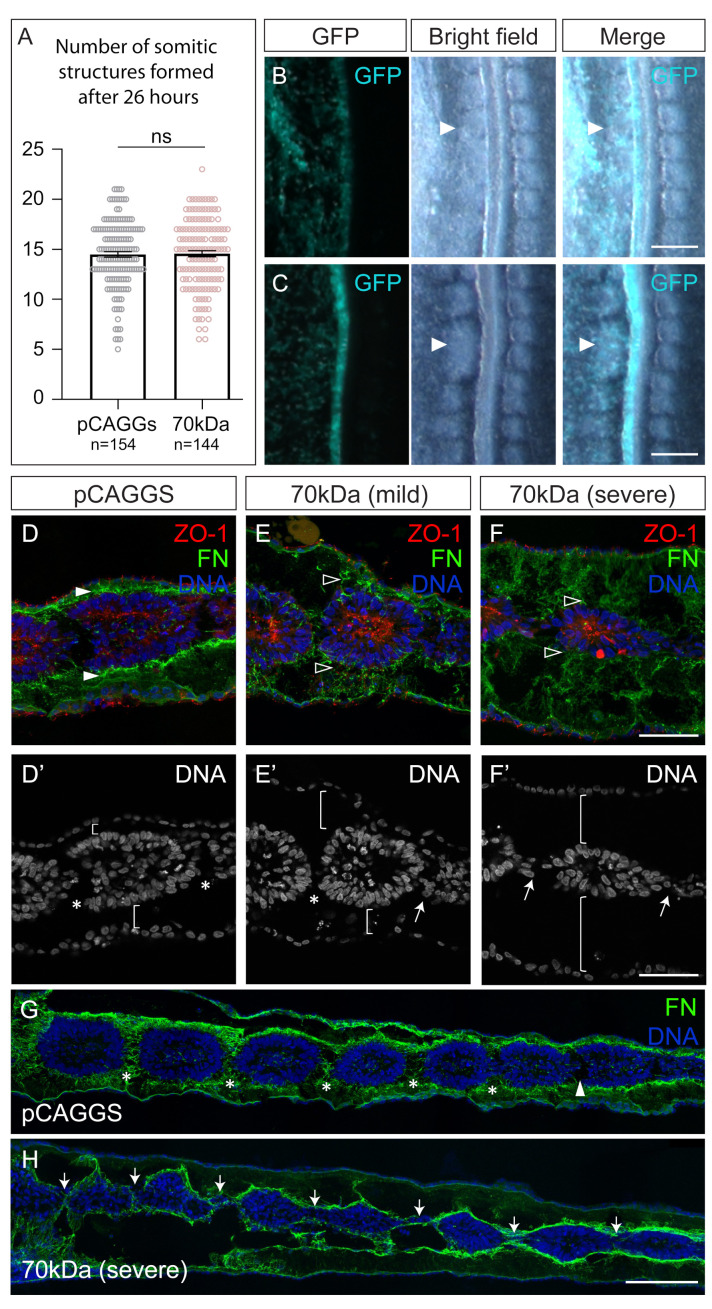
Somite morphology and cleft formation is compromised in 70 kDa-electroporated embryos. (**A**) Number of somitic structures formed in pCAGGs- (mean = 14.5; standard error of the mean (SEM) = 0.27) and 70 kDa-electroporated (mean = 14.6; SEM = 0.29) embryos after 26 h. (**B**,**C**) Close up of embryos electroporated with 70 kDa showing examples of fewer somites (**B**), ill-defined somites (arrowheads in **B**) and fused somitic structures (arrowheads in **C**) on the electroporated side (GFP, left). (**D**–**F**) Sagittal views of somite SI in embryos electroporated with pCAGGs (**D**,**D’**) and 70 kDa with either mild (**E**,**E’**) or severe (**F**,**F’**) phenotypes, immunostained for ZO-1 and fibronectin and stained for DNA. SI of pCAGGs- and 70 kDa-electroporated embryos polarize ZO-1 normally (**D**–**F**). A thick fibronectin matrix surrounds the somites of pCAGGs-electroporated embryos (white arrowheads in **D**), while this matrix is disrupted in 70 kDa-electroporated embryos (open arrowheads in **E**,**F**). Somitic clefts (*) form in pCAGGs embryos, whereas 70 kDa-electroporated embryos retain cells within one of the clefts (arrow in **E’**) or in both (arrows in **F’**) clefts, suggesting incomplete cleft formation. Somites of 70 kDa-electroporated embryos are also detached from either the endoderm or ectoderm (brackets in **E’**), or from both (brackets in **F’**). (**G**,**H**) Sagittal sections of embryos electroporated with pCAGGs and 70 kDa (severe phenotype). Asterisks mark complete clefts, filled with a fibronectin matrix, and arrowhead indicates forming cleft (**G**). Arrows point to incomplete somitic clefts (**H**). Rostral to the left and dorsal on top. Error bars indicate standard error of the mean. ns—not statistically significant (paired Student’s *t*-test). GFP—green fluorescent protein. FN—fibronectin. ZO-1—Zonula occludens protein 1. Scale bars: 200 µm (**B**,**C**), 50 µm (**D**–**F’**), 100 µm (**G**,**H**).

**Figure 3 cells-11-02003-f003:**
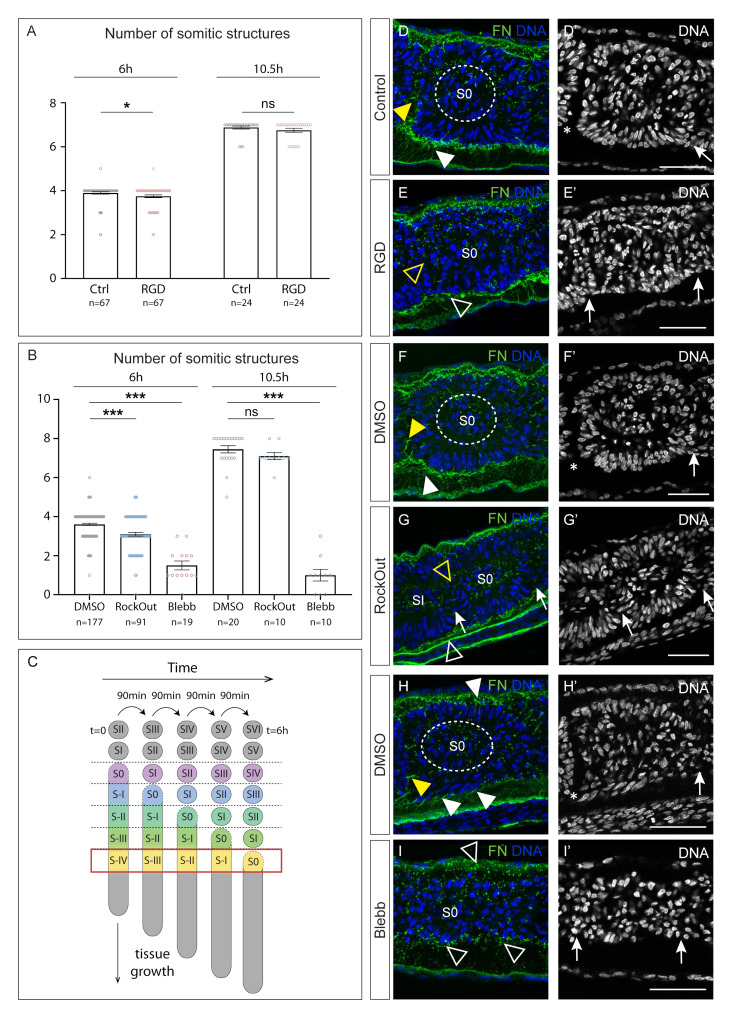
Somite morphology and cleft formation are affected by RGD, RockOut and Blebbistatin. (**A**) Number of somitic structures formed after 6 or 10.5 h in culture in control (mean = 3.9; SEM = 0.04; mean = 6.9; SEM = 0.09) and RGD-treated (mean = 3.7; SEM = 0.07; mean = 6.8; SEM = 0.09) explants. (**B**) Number of somite structures formed in explants cultured for 6 or 10.5 h with DMSO (mean = 3.6; SEM = 0.05; mean = 7.5; SEM = 0.19), RockOut (mean = 3.1; SEM = 0.09; mean = 7.1; SEM = 0.18) or Blebbistatin (mean = 1.5; SEM = 0.23; mean = 1.0; SEM = 0.30). (**C**) Strategy of analysis: Here we analysed the S0 region after 6 h in culture, which corresponds to S-IV when the treatments began (red box). (**D**–**I’**) Sagittal views of explant pairs in experiments aimed at testing the effect of RGD (**D**–**E’**), RockOut (**F**–**G’**) and Blebbistatin (**H**–**I’**) on somite formation. Immunostaining for fibronectin and staining for DNA. Rostral is to the left and dorsal on top. In control explants, the fibronectin matrix around the tissue is fibrillar and thick (white arrowheads in **D**,**F**,**H**). S0 rostral cleft formation is complete in control explants (asterisks in **D’**,**F’**,**H’**) and a fibronectin matrix is detected within this cleft (yellow arrowheads in **D**,**F**,**H**). The fibronectin matrix in RGD- and RockOut-treated explants is thinner than in the contralateral control (open white arrowheads in **E**,**G**). S0 rostral cleft is incomplete (arrow in **E’**,**G’**) and lacks fibronectin (open yellow arrowhead in **E**,**G**). Blebbistatin-treated explants display only a few thin fibres of fibronectin and mostly granular staining, suggesting unassembled fibronectin protein (open arrowheads in **I**). No cleft is formed (arrows in **I’**). The somite-characteristic centripetal alignment of nuclei is perturbed in RGD-treated explants (**E**’), normal in RockOut-explants (**G’**) and absent in Blebbistatin-explants (**I’**). Complete somite clefts are marked with asterisks and incomplete clefts with arrows. Dotted circle indicates the somitocoel around which control cells organize in a centripetal manner. Blebb—Blebbistatin. FN—fibronectin. Error bars in A,B indicate standard error of the mean. *p* values were calculated using a paired Student’s *t*-test. ns—not significant, * *p* < 0.05, *** *p* < 0.001. Scale bars: 50 µm.

**Figure 4 cells-11-02003-f004:**
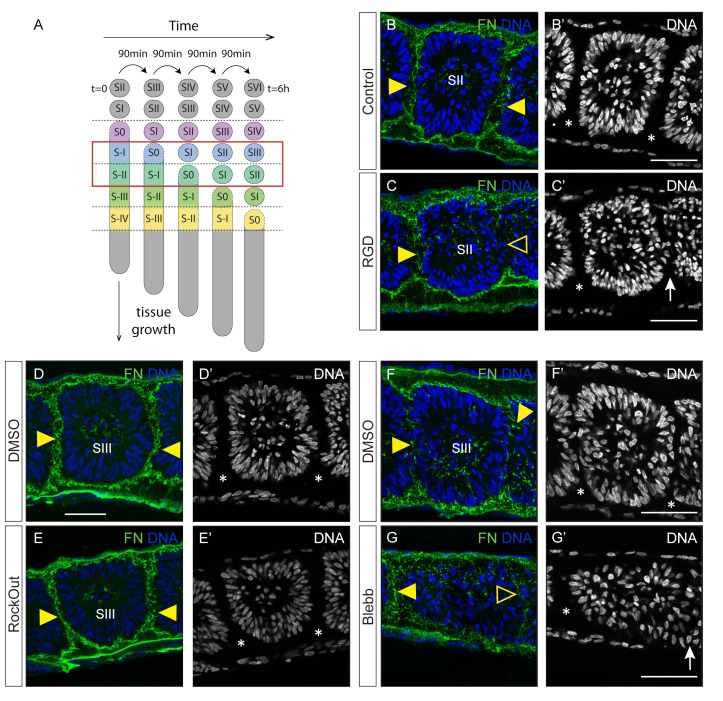
Impact of RGD, RockOut or Blebbistatin treatment on determined PSM segmentation. (**A**) Strategy of analysis: Here we examine the somites SII/SIII after 6 h in culture, which corresponded to stage S-II/S-I when the treatments began (red box). (**B**–**G**) Sagittal views of explants cultured with RGD (**B**–**C’**), RockOut (**D**–**E’**) or Blebbistatin (**F**–**G’**) and their respective controls. Immunostaining for fibronectin and staining for DNA. Rostral is left and dorsal on top. Explant halves cultured in the presence of RGD display incomplete cleft formation caudal to SII (compare **B’**,**C’**), which lacks fibronectin matrix (open yellow arrowhead in **C**). Somite SIII in RockOut-treated explants is indistinguishable from control explants (**D**,**D’**,**E**,**E’**). SIII in Blebbistatin-treated explants have incomplete caudal clefts with a thin fibronectin matrix (yellow arrowhead in **F**, open yellow arrowhead in **G**) and the somite nuclei are not aligned as in the control (**F’**,**G’**). Complete clefts are marked with asterisks and incomplete clefts with arrows. Blebb—Blebbistatin. FN—fibronectin. Scale bars: 50 µm.

**Figure 5 cells-11-02003-f005:**
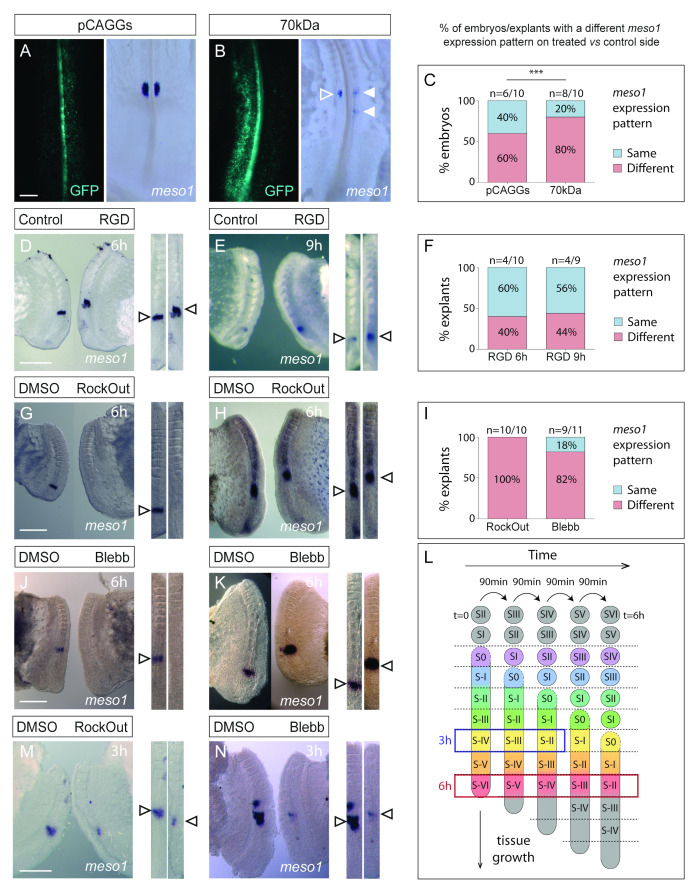
*Meso1* expression is altered in 70 kDa-electroporated embryos and explants cultured with RGD, RockOut or Blebbistatin. In situ hybridization for *meso1* on embryos electroporated with the pCAGGS or 70 kDa (**A**,**B**) and on explant cultures testing the effect of RGD (**D**,**E**), RockOut (**G**,**H**,**M**) or Blebbistatin (**J**,**K**,**N**). Straightened images of the paraxial mesoderm of contralateral explant pairs were aligned by SIV (right, **D**,**E**,**G**,**H**,**J**,**K**,**M**,**N**) to assess whether the *meso1* expression pattern was the same or different on the two contralateral sides. Rostral is on top. Quantification of the percentage of PSM pairs with different expression patterns (**C**,**F**,**I**). Example of a pCAGGs-electroporated control embryo with the same *meso1* expression pattern on both sides (**A**) and a 70 kDa-electroporated embryo where the electroporated side (left) has a different *meso1* pattern from the non-electroporated side (solid vs. open arrowheads in **B**). The percentage of 70 kDa-electroporated embryos with a different *meso1* pattern on the electroporated side is significantly higher than with pCAGGs. *Meso1* expression patterns in RGD-treated explants after 6 (**D**) or 9 h (**E**) of culture are different from their contralateral controls in 40%–44% of cases (**F**). Examples of *meso1* expression patterns in RockOut- (**G**,**H**) and Blebbistatin-treated (**J**,**K**) explants compared to that of their contralateral control (DMSO) sides (arrowheads, **G**,**H**,**J**,**K**). In total, 100% of RockOut-treated and 82% of Blebbistatin-treated explants displayed a different pattern than the contralateral control after 6 h of culture (**I**). RockOut- and Blebbistatin-treated explants incubated for 3 h also produced different *meso1* expression patterns compared to the controls (arrowheads, **M**,**N**). (**L**) Representation of explant analysis in experiments with 3 and 6 h culture periods. *Meso1* is induced in S-II. When the treatment was added 6 h earlier, the perturbation of *meso1* expression occurred between stages S-VI and S-II (red frame). RockOut- and Blebbistatin-treated explants cultured for 3 h also showed altered *meso1* expression indicating that their effect occurred between S-IV and S-II (blue frame). *p* values were calculated using a chi-square test. Blebb—Blebbistatin. *** *p* < 0.001. Scale bars: 200 µm (**A**,**B**), 500 µm (**D**–**N**).

**Figure 6 cells-11-02003-f006:**
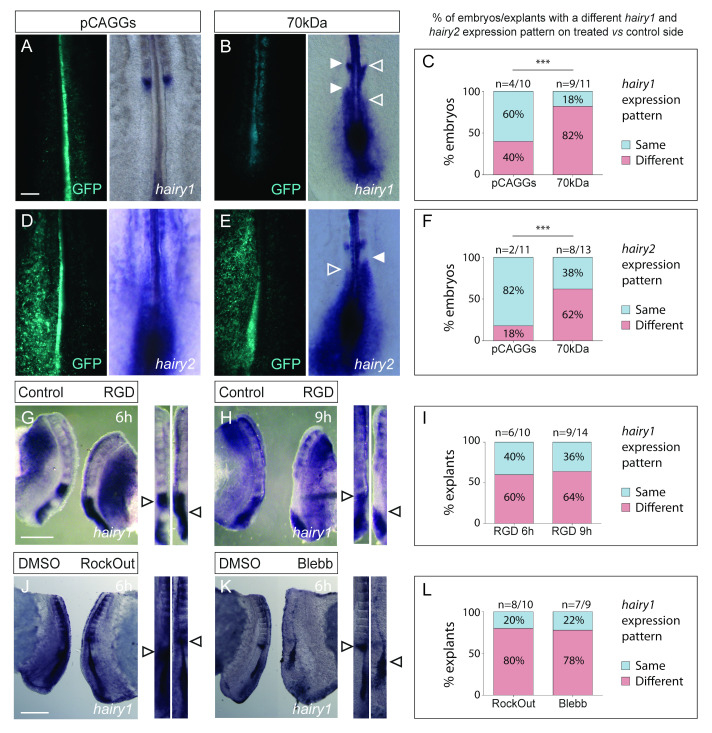
Perturbed segmentation clock dynamics in 70 kDa-electroporated embryos and explants cultured with RGD, RockOut or Blebbistatin, evidenced by altered *hairy1* or *hairy2* expression patterns. In situ hybridization for *hairy1* (**A**,**B**) and *hairy2* (**D**,**E**) on pCAGGS- (**A**,**D**) and 70 kDa-electroporated (**B**,**E**) embryos and on explant pairs testing the effect of RGD (**G**,**H**), RockOut (**J**) or Blebbistatin (**K**). Straightened images of paraxial mesoderm of contralateral explant pairs were aligned by SIV (right, **G**,**H**,**J**,**K**) to determine if the phase of the *hairy1* expression cycle was the same in the two contralateral sides. Rostral is on top. Quantification of the percentage of PSM pairs which are in a different phase of *hairy1/2* expression (**C**,**F**,**I**,**L**). Example of a pCAGGs-electroporated control with the same *hairy1* (**A**) and *hairy2* (**D**) expression pattern on both sides of the PSM. The 70 kDa-electroporated embryos with different *hairy1* (**B**) and *hairy2* (**E**) expression patterns on the electroporated sides (left) (solid vs open arrowheads in **B**,**E**). The percentage of 70 kDa-electroporated embryos with altered *hairy1* (**C**) and *hairy2* (**F**) expression patterns is significantly higher than in controls (pCAGGs) (**C**,**F**). Examples of *hairy1* expression patterns in RGD-treated explants after 6 (**G**) or 9 h (**H**) of culture. In total, 60%–64% of RGD-treated explants display different *hairy1* patterns than their contralateral control sides (arrowheads in **G**,**H**,**I**). In RockOut- (**J**) and Blebbistatin-treated (**K**) explants, 80% and 78%, respectively, displayed different *hairy1* expression compared to their contralateral control (DMSO) explants after 6 h of culture (arrowheads in **J**,**K**,**L**). *p* values were calculated using a chi-square test. Blebb—Blebbistatin. *** *p* < 0.001. Scale bars: 200 µm (**A**–**E**), 500 µm (**G**–**K**).

**Figure 7 cells-11-02003-f007:**
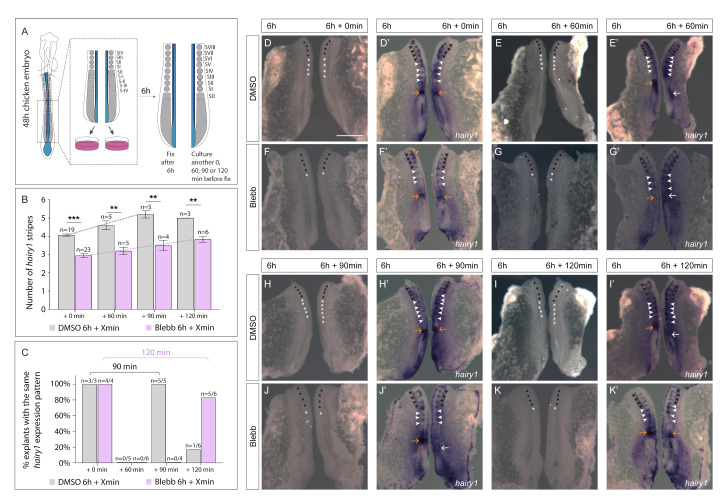
*hairy1* expression oscillates with a periodicity of 120 min in explants cultured with Blebbistatin. (**A**) Experimental setup: posterior embryo halves were cultured separately in the same conditions. After 6 h, one of the explants was fixed and the other one was either fixed at the same time (0 min) or cultured for another 60, 90 or 120 min. (**B**) Number of *hairy1* expression stripes observed after different incubation periods. Explants cultured with Blebbistatin form significantly fewer *hairy1* expression stripes. (**C**) Percentage of explant pairs showing the same *hairy1* expression pattern in different time intervals. (**D**–**K’**) Dorsal views of representative examples of explant pairs cultured in DMSO (**D**–**E’**,**H**–**I’**) or Blebbistatin (**F**–**G’**,**J**–**K’**). Each explant pair was imaged twice: prior to *hairy1* probe addition (**D**–**K**) and after in situ hybridization staining (**D’**–**K’**). Black and white asterisks indicate visible somitic structures before and after culture, respectively. Black arrowheads mark molecular segments characterized by *hairy1* expression stripes formed before culture, white arrowheads are stripes of *hairy1* expression formed during culture. Same colour arrows indicate the same *hairy1* expression pattern in contralateral PSMs, while arrows with different colours indicate different expression patterns. Rostral to the top. *p* values were calculated using a Kruskal–Wallis test. Error bars in B indicate standard error of the mean. See also: data in Appendix A. Blebb—Blebbistatin. *** *p* < 0.001; ** *p* < 0.01. Scale bars: 500 µm.

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
