# Peer review of "Cell–Fibronectin Interactions and Actomyosin Contractility Regulate the Segmentation Clock and Spatio-Temporal Somite Cleft Formation during Chick Embryo Somitogenesis"

_cells, 2022, doi:10.3390/cells11132003_

Round 1

Reviewer 1 Report

This paper by Gomes de Almeida and colleagues investigates the relationship between fibronectin assembly and somitogenesis at both the morphoplogical level (someite epithelialization) and genetic level (changes in gene expression in time and space as part of the clock).

The initial hypotheis is clear and the experimental strategies are well suited to test it.

The results obtained seem to back up all claims and appropriate controls are provided. Data presentation however has some issues and hinders the impact and interpretation of the data

I suggest some changes:

1/ Statistical significance is reached for some analysis while not for other despite very small differences in actual means and std error/deviation (e.g. fig3). Please indicate in the text the actual mean and error so that one can compare what differences we are looking at. Also consider using a better way of plotting data such as box and whiskers instead of bars which do not allow for assessment of distribution of data points. When counting number of somites I suspect that the authors ended up with embryos in which there were 4 somites and other with only 3 (fig.3A RGD vs control). In this case means and bar plots are irrelevant. What matters is the change of proportion between embryos with 4 s vs 3s. Do we go from 50%/50% to 25/75 or else.... And statistical tests for proportions between samples should be used rather than t-tests. Also here, defining somites while acknowledging that they do not properly form is strange. Why count 3 or 4 somites if none of them has an actual boundary. The way of counting and defining somites may be too conservative and may hide the actual phenotype which is looking much more severe on the images than on the graph supposedly summarizing the results. This dichotomy between images and graphs needs to be addressed throughout the paper to better represent the results of each experiments.

2/ Qualitative definition such as detached vs severly detached (supp fig2) add nothing but confusion. Either define precisely what criteria is used to separate such conditions or merge them. Some logic should be applied when qualitative categories are mentionned > define based on a quantitative criteria that is used as a threshold or merge categories that are alike.

3/ Presentation of data on Fig5 needs to be improved. What is shown in panel F for instance? 6h vs 9h of what controls? RGD? both? What is what? Instead of having altered vs normal the authors could plot the position of the region of expression as a function of distance from a reference somite and plot a graph for each experimental conditions. This would allow to visualize the shift in spatial distribution and the variability (one region of expression vs two). The categories normal vs altered are based on the authors' gut feeling. Same logic applies to Fig6.

Author Response

Reviewer 1: This paper by Gomes de Almeida and colleagues investigates the relationship between fibronectin assembly and somitogenesis at both the morphological level (somite epithelialization) and genetic level (changes in gene expression in time and space as part of the clock).

The initial hypothesis is clear and the experimental strategies are well suited to test it.

The results obtained seem to back up all claims and appropriate controls are provided. Data presentation however has some issues and hinders the impact and interpretation of the data

I suggest some changes:

1/ Statistical significance is reached for some analysis while not for other despite very small differences in actual means and std error/deviation (e.g. fig3). Please indicate in the text the actual mean and error so that one can compare what differences we are looking at. Also consider using a better way of plotting data such as box and whiskers instead of bars which do not allow for assessment of distribution of data points. When counting number of somites I suspect that the authors ended up with embryos in which there were 4 somites and other with only 3 (fig.3A RGD vs control). In this case means and bar plots are irrelevant. What matters is the change of proportion between embryos with 4 s vs 3s. Do we go from 50%/50% to 25/75 or else… And statistical tests for proportions between samples should be used rather than t-tests. Also here, defining somites while acknowledging that they do not properly form is strange. Why count 3 or 4 somites if none of them has an actual boundary. The way of counting and defining somites may be too conservative and may hide the actual phenotype which is looking much more severe on the images than on the graph supposedly summarizing the results. This dichotomy between images and graphs needs to be addressed throughout the paper to better represent the results of each experiment.

REPLY: We appreciate the comments received that have aided in significantly improving our manuscript. We have heeded Revewer1’s advice and added the mean and standard error in the text for best comparison of the data. Also, we changed data representation in Figures 2A, 3A and 3B to include the raw data, which allows an appreciation of the distribution of the data points obtained. We chose not to represent the proportion between embryos with 4s vs 3s, as suggested, because in many cases the somites are ill-formed, so a direct comparison would also be a mis-representation of the phenotypes obtained.

Reviewer1 is right in that we took a conservative approach to count the number of somitic structures (new term employed in Figures 2A, 3A and 3B) formed in each condition. We recognize this was not sufficiently clear in the manuscript, which may explain the referred apparent dichotomy between images and graphs. We chose to count a “somite” whenever a boundary was visible in the dissecting microscope. However, in contrast to the somitic structures formed in control conditions, where these boundaries are always very clear, in our treatments the clefts were poorly formed, encompassing what we designated as “ill-defined somites”. We have now included a clarification of this term in the main text, along with a new Supplementary Figure S3 that (together with Figure 2B,C) illustrates what is considered “ill-defined somites”.

2/ Qualitative definition such as detached vs severly detached (supp fig2) add nothing but confusion. Either define precisely what criteria is used to separate such conditions or merge them. Some logic should be applied when qualitative categories are mentionned > define based on a quantitative criteria that is used as a threshold or merge categories that are alike.

REPLY: We agree with the comments received and have merged the categories “detached” and “severely detached tissues” in the Figure. We have also included “fewer somites” within the “somite phenotypes” category.

3/ Presentation of data on Fig5 needs to be improved. What is shown in panel F for instance? 6h vs 9h of what controls? RGD? both? What is what? Instead of having altered vs normal the authors could plot the position of the region of expression as a function of distance from a reference somite and plot a graph for each experimental conditions. This would allow to visualize the shift in spatial distribution and the variability (one region of expression vs two). The categories normal vs altered are based on the authors' gut feeling. Same logic applies to Fig6.

REPLY: We thank Reviewer1 for identifying difficulties in interpretation within Figures 5 and 6, which we have now addressed. We have added proper labelling to the panels and also changed the terminology “normal/altered” to “same/different”, since the former suggested the use of subjective criteria in explant classification. The methodology we used was a direct comparison between the expression patterns obtained in contralateral control vs treated explants. Explant images were straightened using the Straightened tool in Fiji and aligned rostrally by the caudal cleft of SIV. We compared meso1 mexpression patterns by observing the number of bands present, their relative intensity and A-P position within the PSM. Note that since meso1 is upregulated in S-II, stays expressed until S-I and is then downregulated there and upregulated again in S-II, both A-P position and the intensity of the band reflect differences in the meso1 expression cycle. For scoring hairy1/2 expression, we used the three described phases of the clock cycle (Palmeirim et al., 1997) as a guideline to assess whether the pattern was the “same” or “different” in the experimental pairs. We have included a more complete explanation of these terms and how they were assessed in the Materials and Methods and in the figure legends.

Reviewer 2 Report

Title: Cell-fibronectin interactions and actomyosin contractility regulate the segmentation clock and spatio-temporal somite cleft formation during chick embryo somitogenesis

Authors: Patrícia Gomes de Almeida, Pedro Rifes, Ana P. Martins-Jesus, Gonçalo G. Pinheiro, Raquel P. Andrade, and Sólveig Thorsteinsdóttir

In this manuscript, the authors study the influence of fibronectin on somite formation. Through a number of different approaches, the authors show that somite formation is dependent on the fibronectin matrix itself, as well as integrin-fibronectin binding and downstream molecular processes. The authors have some very interesting findings presented in exceptional figures, especially the link between cellular mechanisms and segmentation periodicity genes. The manuscript appears to be scientifically sound and in need of little in the way of modification. Below are some of my minor suggestions that stand to improve the manuscript and the study described therein (in no particular order of importance).

1. Lines 85-86, the authors write “First, we experimentally perturbed extracellular fibronectin fibrillogenesis or inhibited integrin-fibronectin binding through RGD” - I would suggest changing “or” to “and”. The way it is written now sounds like they did one or the other, when in reality, they did both.

2. Line 86. RGD is first mentioned here with no description of what it is or what it stands for. The authors should describe it here.

3. The first part of section 3.1 in the Results section appears to be all description of methods, including Figure 1. The only “results” is the sentence “None of experimental conditions tested lead 202 to an increase in apoptosis (Figure S1).”
The authors should consider moving this text and figure 1 into the Methods section. There don’t appear to be any results here.

4. The authors begin by describing Figure 2D in line 227 and the Figure 2A later in the text. Traditionally the panels are described in order (within reason). Personally, when I was reading and read Figure 2D, I thought I’d missed text about 2A. The authors may want to consider revising this in some way,  if possible. 

5. The authors state that after 70kDa-electroporation, only 10 % of animals had “fewer somites”. Despite being statistically significant, this does not appear to be much.
This draws several questions.

A. Could the fewer somites simply be due to the detachment of tissues, kinked neural tube and other defects and not due to fibronectin matrix disruption directly? After all, 90% of embryos still produce the correct number of somites even after disruption of the fibronectin matrix.

B. Is there a difference between fused somites and “fewer somites”? There authors did not show the number of animals with “fused somites” in Figure S2.

6. Lines 273-274. The authors state “Altogether, these results indicate that an intact fibronectin matrix is required for proper somite cleft formation.”
In how many animals were “incomplete clefts” observed in the 70kDa-electroporation study? Overall the numbers affected seem rather low for something that is so important. It makes one wonder if the general phenotype of the animal being so disrupted affected the clefts or the somites themselves.

7. Could Figure S5 be included as part of the main manuscript in the conclusion section? The authors may want to consider this.

8. Lines 363-364 after discussing treatment with RGD, RockOut and Blebbistatin, the authors state “We conclude that cell-fibronectin interactions and actomyosin contractility are required for somite cleft formation.” Does cell polarization absolutely require fibronectin? Could it be that RGD, RockOut or Blebbistatin affect cell polarization and thereby cause disrupted smite clefts, which in turn don’t allow fibronectin to accumulate between the somites?
Thus, the disrupted clefts not containing fibronecitn may be due to simple lack of space rather than the lack of fibronectin causing the cleft disruption?

8. Lines 375 - 377 The authors state “Nevertheless, there was a significant increase in the frequency of meso1 expression altertions in embryos electroporated with the 70kDa fragment, when compared with controls (p<0.01; Figure 5A-C” Could this be due to the general tissue being disrupted and not somites directly affected per se?  Why or why not?

Author Response

Reviewer 2: In this manuscript, the authors study the influence of fibronectin on somite formation. Through a number of different approaches, the authors show that somite  formation is dependent on the fibronectin matrix itself, as well as integrin-fibronectin binding and downstream molecular processes. The authors have some very interesting findings presented in exceptional figures, especially the link between cellular mechanisms and segmentation periodicity genes. The manuscript appears to be scientifically sound and in need of little in the way of modification. Below are some of my minor suggestions that stand to improve the manuscript and the study described therein (in no particular order of importance).

REPLY: We appreciate the suggestions received from Reviewer2, that have contributed to significantly improving our manuscript.

1. Lines 85-86, the authors write “First, we experimentally perturbed extracellular fibronectin fibrillogenesis or inhibited integrin-fibronectin binding through RGD” - I would suggest changing “or” to “and”. The way it is written now sounds like they did one or the other, when in reality, they did both.

REPLY: Thank you for pointing that out. We have changed the text accordingly.

2. Line 86. RGD is first mentioned here with no description of what it is or what it stands for. The authors should describe it here.

REPLY: Thank you for noticing. We have added a short description to the phrase, which now reads as follows: “…the RGD peptide which competes with fibronectin for integrin binding and also perturbs fibronectin matrix assembly.”

3. The first part of section 3.1 in the Results section appears to be all description of methods, including Figure 1. The only “results” is the sentence “None of experimental conditions tested lead 202 to an increase in apoptosis (Figure S1).”
The authors should consider moving this text and figure 1 into the Methods section. There don’t appear to be any results here.

REPLY: We have moved the first part of section 3.1 and Figure 1 to the Materials and Methods section, with a few minor adjustments.

The Results section now starts as follows: “Four experimental treatments were used to interfere with fibronectin-integrin-actomyosin axis (Figure 1A), none of which led to an increase in apoptosis (Figure S1). To inhibit fibronectin matrix assembly, a plasmid….”.

We agree this is a much better organization.

4. The authors begin by describing Figure 2D in line 227 and the Figure 2A later in the text. Traditionally the panels are described in order (within reason). Personally, when I was reading and read Figure 2D, I thought I’d missed text about 2A. The authors may want to consider revising this in some way, if possible. 

REPLY: We agree that the panels should be described suggested order to avoid confusion and have accordingly moved the text referring to Figure 2D further down. The text now starts by referring to Figure 2A, B etc.

5. The authors state that after 70kDa-electroporation, only 10 % of animals had “fewer somites”. Despite being statistically significant, this does not appear to be much.

REPLY: We want to start by clarifying that we took a conservative approach when counting the number of somitic structures (new term employed in Figures 2A, 3A and 3B) formed in each condition. We recognize this was not sufficiently clear in the manuscript. We chose to count a “somite” whenever a cleft was visible in the dissecting microscope. However, in contrast to the somitic structures formed in control conditions, where these clefts are always very clear, in our treatments the boundaries were poorly formed, encompassing what we designated as “ill-defined somites”. We have now included this clarification in the main text, along with a new Supplementary Figure S3 that (together with Figure 2B,C) illustrates all types of somitic structures accounted for. We have further modified the text to better explain the phenotype mentioned (line 236): “In some cases (15/144) somite-like clefts were no longer discernable”. Furthermore, we also saw other somite phenotypes (irregularly shaped somites, fused somites, fragmented somites, misaligned somites) which are shown in Figure S2. In this version of the manuscript we included “fewer somites” into the “somite phenotypes” category.

This draws several questions.

A. Could the fewer somites simply be due to the detachment of tissues, kinked neural tube and other defects and not due to fibronectin matrix disruption directly? After all, 90% of embryos still produce the correct number of somites even after disruption of the fibronectin matrix.

REPLY: We believe this is not the case. Of the 110 embryos with detached tissues, only 35 had what we categorize as somite phenotypes. Conversely, the absence of somitic clefts (fewer somites) was also obtained in intact embryos (see Figure 2B). Therefore, we believe that the detachment of the tissues is not the cause of the somite phenotypes observed.

  1. Is there a difference between fused somites and “fewer somites”? There authors did not show the number of animals with “fused somites” in Figure S2.

REPLY: We have now used the term “fused somitic structures” to clarify this phenotype. In this case, we counted two somitic structures.

6. Lines 273-274. The authors state “Altogether, these results indicate that an intact fibronectin matrix is required for proper somite cleft formation.”
In how many animals were “incomplete clefts” observed in the 70kDa-electroporation study? Overall the numbers affected seem rather low for something that is so important. It makes one wonder if the general phenotype of the animal being so disrupted affected the clefts or the somites themselves.

REPLY: Incomplete clefts were observed in all 70kDa-electroporated embryo sides. As stated above, these are referred to ill-formed somitic structures throughout the text, and were not included in the quantification of the other phenotypes observed (Figure S2), unless they were also fused, fragmented, irregularly shaped, etc.  

7. Could Figure S5 be included as part of the main manuscript in the conclusion section? The authors may want to consider this.

REPLY: We thank the reviewer for this suggestion. We made the exercise to move it to the first section of the Discussion. However, we included this summary-type figure to help the reader visualize the results of Figures 3 and 4 and by moving it to the discussion, we find that it does not accomplish this purpose.

8. Lines 363-364 after discussing treatment with RGD, RockOut and Blebbistatin, the authors state “We conclude that cell-fibronectin interactions and actomyosin contractility are required for somite cleft formation.” Does cell polarization absolutely require fibronectin? Could it be that RGD, RockOut or Blebbistatin affect cell polarization and thereby cause disrupted somite clefts, which in turn don’t allow fibronectin to accumulate between the somites? Thus, the disrupted clefts not containing fibronecitn may be due to simple lack of space rather than the lack of fibronectin causing the cleft disruption?

REPLY: Note that perturbing fibronectin matrix assembly with the 70kDa fragment does not block cell polarization (Figure 2 E,E’,F,F’), however somite clefts are not properly formed. Moreover, RockOut treated explants lacked assembled fibronectin in the clefts, despite having normal cell polarization (Figure 3 G,G’). So our results indicate that the lack of fibronectin deposition in the cleft is not a consequence of improper cell polarization, as Reviewer2 suggests. On the contrary, our results suggest that the alpha5beta1 integrin needs to engage with fibronectin because cell polarization is impaired when we add RGD. The following text in the beginning of the Discussion is meant to explain this issue:

“The presence of RGD hampered cell polarization in recently formed somites, while perturbing fibronectin matrix assembly with the 70kDa-construct did not. The reason for this difference is not clear, but while both the 70kDa construct and addition of RGD perturb fibronectin assembly, as evidenced by these embryos displaying a thinner fibronectin matrix (Figures 2E,F and 3E) compared to controls, only RGD blocks integrin-fibronectin binding. Hence, cell polarization during somite emergence may be more sensitive to integrin engagement with the surrounding fibronectin matrix than to its assembly state.” (lines 555-561)

8. Lines 375 - 377 The authors state “Nevertheless, there was a significant increase in the frequency of meso1 expression alterations in embryos electroporated with the 70kDa fragment, when compared with controls (p<0.01; Figure 5A-C” Could this be due to the general tissue being disrupted and not somites directly affected per se?  Why or why not?

REPLY: We don’t think this is the case, because the embryos we processed for in situ hybridization were all intact (i.e. did not have detached tissues) and they still had a different meso1 expression pattern on the electroporated side in the majority of the cases (see Figure 5A-C).

We have included this information in the Materials and Methods section: “Only intact 70kDa-electroporated embryos (without detached tissues) were processed for in situ hybridization.”

Round 2

Reviewer 1 Report

Thank you for revising the manuscript. I support its publication in the present form.